# Probing into the Fine-grained Manifestation in Multi-modal Image Synthesis

## Abstract

The ever-growing development of multi-modal image synthesis brings unprecedented realism to generation tasks. In practice, it is straightforward to judge the visual quality and reality of an image. However, it is labor-consuming to verify the correctness of semantic consistency in the auto-generation, which requires a comprehensive understanding and mapping of different modalities. The results of existing models are sorted and displayed largely relying on the global visual-text similarity. However, this coarse-grained approach does not capture the fine-grained semantic alignment between image regions and text spans. To address this issue, we first present a new method to evaluate the cross-modal consistency by inspecting the decomposed semantic concepts. We then introduce a new metric, called MIS-Score, which is designed to measure the fine-grained semantic alignment between a prompt and its generation quantitatively. Moreover, we have also developed an automated robustness testing technique with referential transforms to test and measure the robustness of multi-modal synthesis models. We have conducted comprehensive experiments to evaluate the performance of recent popular models for text-to-image generation. Our study demonstrates that the proposed metric MIS-Score represents better evaluation criteria than existing coarse-grained ones (*e.g.*, CLIP) to understand the semantic consistency of the synthesized results. Our robustness testing method also proves the existence of biases embedded in the models, hence uncovering their limitations in real applications.

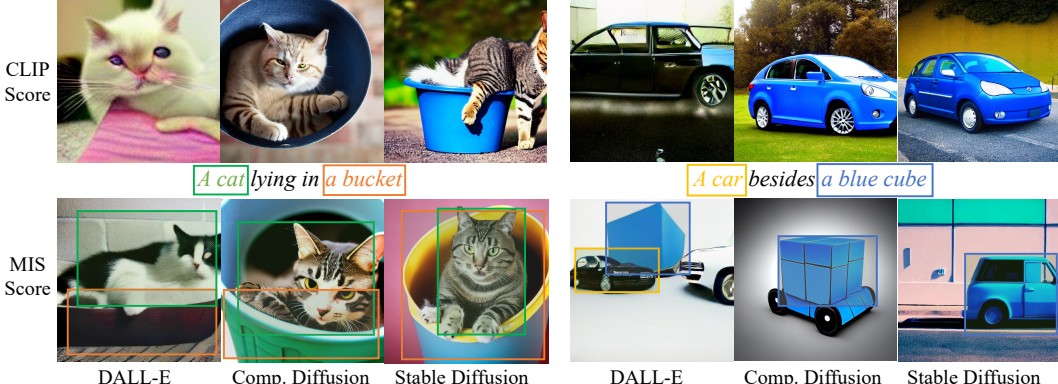

Figure 1: The generated images of state-of-the-art multi-modal image synthesis models (DALL-E (Cho et al., 2022), Composable Diffusion (Liu et al., 2022), Stable Diffusion (Rombach et al., 2022)) selected by CLIP score and MIS-Score for two prompts: (i) *A cat lying in a bucket.* (ii) *A car besides a blue cube.* The green and orange bounding boxes in the prompts and the images indicate the fine-grained alignment of semantic concepts.

# 1   INTRODUCTION

Multi-modal image synthesis (Esser et al., 2021; Ramesh et al., 2021; Rombach et al., 2022) aims to generate images given input prompts such as natural language descriptions or a set of keywords. This multi-modal synthesis has a variety of potential applications, *e.g.*, computer-aided design, text-guided photo editing, *etc*. In recent few years, it has witnessed unprecedented development based on the advance of deep learning techniques like Generative Adversarial Networks (GANs) (Goodfellow et al., 2020; Zhang et al., 2017; Tao et al., 2020) and cross-modal pre-training models, *e.g.*, CLIP (Radford et al., 2021; Ramesh et al., 2021). Compared to the visual clues (segmentation maps, regional edges, *etc.*) adopted for single-modal image synthesis, cross-modal guidance, *e.g.*, language descriptions, provides an alternative but more intuitive and natural ways to express semantic concepts. The flexibility of text-to-image generation greatly lower the barrier for a wide range of public users to unleash their creativity in image generation and editing.

However, due to the substantial domain gap, effective transfer and fusion of heterogeneous information from different modalities remains a big challenge in the multi-modal synthesis tasks. Moreover, there usually exist many-to-many mapping relationships in the multi-modal image synthesis tasks, *i.e.*, one image may correspond to multiple textual descriptions counterparts and vice versa. In practice, the synthesized images are not always consistent with the text prompts given by users. Additional effort is usually required to manually validate and select the satisfying synthesized results. Various aspects of the synthesized images need to be qualified, *e.g.*, visual quality and the accuracy of objects, attributes, and contextual information.Thus, an automatic evaluation process and a comprehensive and objective evaluation metric are of vital importance in assessing the effectiveness of multi-modal image synthesis models.

To achieve these goals, previous works usually report multiple metrics to cover different aspects. The primary factors considered in the evaluation process are image quality and text-image similarity. Popular metrics include Inception Score (IS) (Salimans et al., 2016) and Frechet Inception Distance (FID) (Heusel et al., 2017) for assessing the image fidelity, and R-precision(RP) (Xu et al., 2018) and CLIP score (Radford et al., 2021) for measuring the cross-modal alignment. These metrics work well for the generation from simple prompts, *e.g.,* description of a single object. However, for prompts with multiple objects and additional context information, simply adopting these metrics is insufficient and may lead to inaccurate or inconsistent results. As shown in the first row of Fig. 1, the ranking of the synthesized images based on one of the most common-used metrics (*i.e.*, CLIP score) is not strongly correlated to the language descriptions. Additionally, the existing metrics lack insights in the assessment of fine-grained semantic concepts, such as object existence, attribute accuracy, spatial location, *etc*. These factors are critical in evaluating the performance of multi-modal image synthesis methods, especially when input prompts are composed of multiple semantic concepts.

To address the above issues, we introduce MIS-Score, a new metric for **M**ulti-modal **I**mage **S**ynthesis to measure the cross-modal semantic consistency by inspecting and capturing the fine-grained mapping across the input language and output vision modalities. As shown in Fig. 1, given a prompt *"A cat lying in a bucket"*, we first parse the semantic concepts in the language description as ("a cat", "a bucket"). We then perform visual grounding by locating each semantic concept with the visual components in the generated image to calculate the fine-grained text-image alignment. These semantic concepts can include but not limited to: *subject*, *object*, *location*, *attribute*, and *relationship*. The cross-modal consistency is measured with MIS-Score by aggregating the alignment score on each semantic concept. Based on the proposed metric, we develop an automatic testing technique, referential transform, to evaluate the robustness of models with different combinations of visual concepts. The key idea is that an accurate multi-modal synthesis model should produce consistent generations given prompts with similar meanings. Correspondingly, models should produce different results given prompts with different semantic concepts. And when there is any mutation in the prompt, the synthesis result is required to always be consistent with the input. In this way, the robustness of models can be evaluated and measured by MIS-Score.

Our major contributions in this work are summarized as follows,

- To the best of our knowledge, we, for the first time, propose a new approach for measuring the fine-grained semantic consistency for the multi-modal image synthesis tasks.

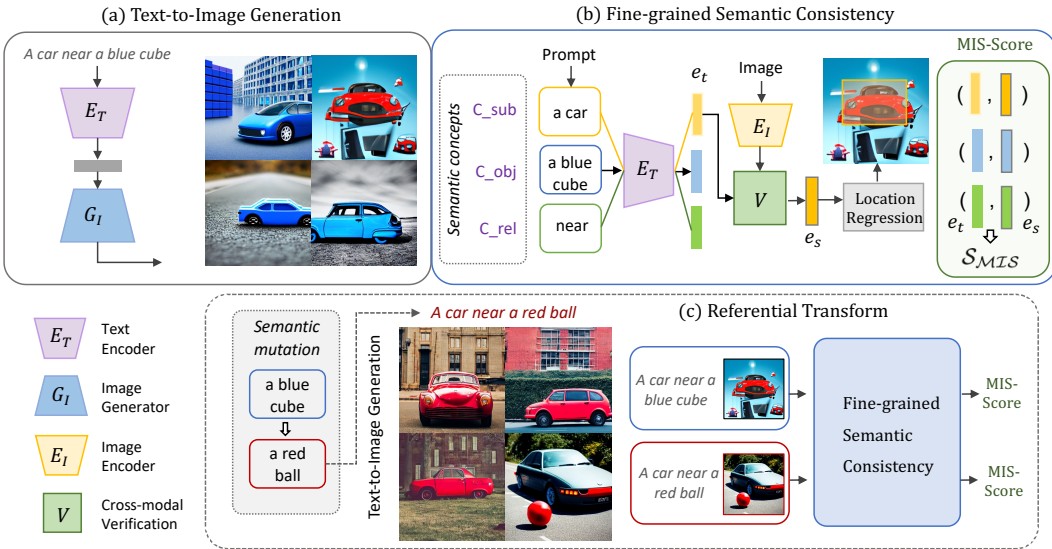

Figure 2: (a) Text-to-Image Generation: Given an input prompt, the results are generated by first encoding the text embeddings and then being transformed into a 3-dimension RGB image. (b) Fine-grained Semantic Consistency: It first parses the prompt into semantic concepts for inspection. Each term is first encoded into embeddings and then grounded in the image for the corresponding visual components. The cross-modal consistency is measured with MIS-Score $\mathcal{S}_{\mathcal{MIS}}$ by aggregating the verification score of each semantic concept. (c) Referential Transform: We first execute a mutation on a selected semantic concept and then apply the image synthesis with the mutated prompt. The robustness of generation models can be measured by calculating the MIS-Score.

- A new metric, MIS-Score, is developed for evaluating the cross-modal alignment, which outperforms the existing coarse-grained ranking mechanism for the synthesized images.
- A robustness testing technique is designed and implemented to evaluate the robustness of text-to-image generation models.

## 2 METHOD

In this section, a new method is proposed for automatically evaluating the performance of multi-modal image synthesis models. Different from metrics comparing the global representation of each modality, we aim to measure the manifestation of fine-grained semantics in the generation. The cross-modal semantic consistency is assessed by inspecting the fine-grained terms from the input prompts. Accordingly, we design a new metric MIS-Score, denoted as $\mathcal{S}_{\mathcal{MIS}}$, which is the aggregation of the cross-modal consistency on all the semantic concepts. The framework of our method is illustrated in Fig. 2. In the following, we first introduce the technique for text-to-image generation. The detailed explanation of MIS-Score can be found in Sec. 2.2 and the robustness testing procedure with referential transform is discussed in Sec. 2.3.

### 2.1 TRANSFERABLE VISUAL MODELS

A standard image model jointly trains an image feature extractor and a decoder to predict the target modality. As a typical example, CLIP proposed by (Radford et al., 2021) jointly trains an image encoder and a text encoder to predict the correct pairings of a batch of (image, text) training examples. During the testing time, the learned text encoder synthesizes a zero-shot linear classifier by embedding the names or descriptions of the target as labels. Inspired by this, recent text-to-image models (Ramesh et al., 2021; Liu et al., 2022; Rombach et al., 2022) aim to train a transformer (Vaswani et al., 2017) to model the text and image embeddings as a single stream of data. They first train an image auto-encoder and then train an auto-regressive transformer to model the joint distribution over

the text and image tokens. In (Ramesh et al., 2021), the overall learning objective is to maximize the evidence lower bound (ELB) on the joint likelihood of the model distribution over images $x$, captions $y$, and the tokens $z$ for the encoded images:

$$
\begin{aligned}
\ln p_{\theta,\psi}(x,y) \geqslant \mathop{\mathbb{E}}_{z \sim q_\phi(z|x)} (\ln p_\theta(x \mid y, z) - \\
\beta D_{\mathrm{KL}} \left( q_\phi(y, z \mid x), p_\psi(y, z) \right)).
\end{aligned}
\tag{1}
$$

However, it is widely discussed that current models cannot effectively handle long prompts with various compositions of concepts (Saharia et al., 2022; Cho et al., 2022; Liu et al., 2022). Models tend to concentrate on the salient component (*e.g.*, the main object in the description), and overlook other contextual details. It also explains why models often fail in precise generation given prompts with a combination of objects.

## 2.2 FINE-GRAINED CROSS-MODAL CONSISTENCY

**Fine-grained semantic concepts.** A single image usually contains rich semantics, covering multiple objects with attributes and/or the description of their context. It is costly to annotate all these semantic concepts with different granularity in large-scale image-text datasets. To this end, we use sentence constituent parsing (Mitchell, 1994) to extract the semantic concepts in each prompt. As shown in Fig. 2, we choose to collect a pool of concepts including *subject*, *object*, and *relation*, by using off-the-shelf language parser (Schuster et al., 2015). Given the concept pool, the semantic representations for regions are created in two steps: 1) We extract a short phrase for each concept by grouping the words in the children of each target word. For example, the *object* phrase in "a car near a blue cube" will be parsed as "*a blue cube*". 2) Each phrase is then encoded into semantic representations by using the pretrained language encoder $E_T$. $E_T$ is a language encoder that converts a natural language text to a semantic representation. Finally, all regional concepts are represented by their semantic embeddings $\left\{ e_t^i \right\}_{i=1,\ldots,C}$, $C$ denotes the size of concept pool. While the semantic concepts are parsed from image descriptions, our method is not constrained by the particular texts that pair with images. More importantly, in light of the powerful language encoder which has seen many words in natural language, we can easily expand our concept pool and scale it up by collecting more specific attributes, like color and gender, which is difficult to achieve by human annotations.

**Visual component validation.** Each semantic embedding $e_t \in \mathcal{R}_{C \times L}$ encoded by $E_T$ is then located in the generated image to its corresponding visual features. Our goal is to measure the semantic concept in a regional visual-semantic space, which covers rich object concepts that can be used for measuring the semantic alignment at scale. In the visual-semantic space, the visual regional embeddings $e_v \in \mathcal{R}_{C \times H \times W}$ extracted from image representations should be matched to text representation $e_t$. $E_I$ is a visual encoder that takes image $I$ and a region location and outputs a visual representation for this region.

Here, the semantic embeddings $e_t$ serve as the query and the visual embeddings $e_v$ act as the key and value. With the multi-head attention, the related visual features will be gathered for each semantic embedding. The gathered semantic features are organized as a semantic map $e_s \in \mathcal{R}_{C \times H \times W}$ that is spatially aligned with the visual feature map $e_v$. Thereafter, we project both the feature maps $e_t$ and $e_s$ to the same semantic space (via linear projection and L2 normalization) obtaining $e_t'$ and $e_s'$, and compute their semantic correlation as the consistency score for each pair $(e_t, e_s)$ as:

$$
\mathcal{F}(e_t, e_s) = \alpha \cdot \exp\left( -\frac{(1 - e_t' e_s')^2}{2\sigma^2} \right),
\tag{2}
$$

where $\alpha$ and $\sigma$ are learnable parameters. The verification scores model the semantic relevance of each visual feature to the linguistic expression. Therefore, we can establish a more salient feature map for the referred semantic concept by modulating the visual features with the verification scores pixel-wisely. Last, the cross-modal consistency between the input prompt and the generated image is calculated by aggregating the score for each semantic concept re-weighted by $w_i$:

$$
\mathcal{S}_{\mathcal{MIS}} = \sum_{i=0}^{C} w_i \mathcal{F}(e_v^i, e_t^i).
\tag{3}
$$

## 2.3 ROBUSTNESS TESTING WITH REFERENTIAL TRANSFORM

As mentioned in (Rombach et al., 2022; Cho et al., 2022), current models tend to encode biases in the training dataset, which largely limits the generality and robustness of the generation. Referential transparency (Søndergaard & Sestoft, 1990; Mitchell, 2002) has been adopted as a key property of programming languages. An expression is "referentially transparent" if replacing a term in that expression with another term that refers to the same value does not alter the semantic meaning. If the term is replaced by another term with a different value, the semantic meaning of the expression should be altered.

Inspired by referential transparency, we propose referential transform, an automated semantic consistency checking technique for testing the robustness of text-to-image generation models. It is a one-way evaluation method without referring back to the source modality. The general idea is that a perfect multi-modal synthesis should produce alike generations given similar prompts, while producing different results when the semantic concept changes. For a referred semantic concept, a referential transform is to change it to another semantic concept. For example, in Fig. 3, given "*A little bird on a tree*", we change the relationship between "*little bird*" and "*tree*" from "on" to "under". It is discovered in our study that the existing models often struggle in generating precise images under certain combinations of semantic concepts.

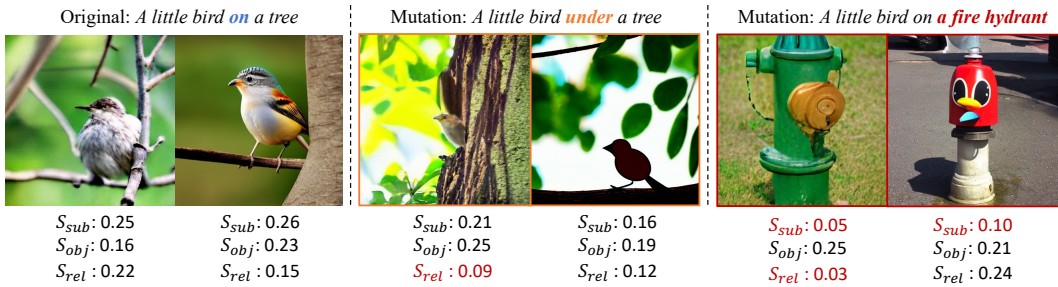

Figure 3: Demonstration of robustness testing with referential transforms: Given "A little bird on a tree" as the input prompt, we first parse the semantic concepts in it. Mutation for *relation* term: change "on" → "under" and Mutation for *object* term: "tree" → "fire hydrant".

Our robustness testing procedure is as follows: (i) Given an input prompt $t_o$, we first parse the phrase of the semantic concept for inspection in the following step. (ii) We then select another semantic concept to replace the original one in the language description, yielding a mutated prompt $t_m$. (iii) we generate new images with $t_m$ and then check the cross-modal consistency using $\mathcal{S}_{\mathcal{MIS}}$ between the newly synthesized images and the mutated prompt. An accurate multi-modal synthesis model should produce similar MIS-Scores given similar prompts, while producing different results given different prompts.

## 3 EXPERIMENTS

### 3.1 IMPLEMENTATION

**Multi-modal image synthesis.** In this section, we conduct experiments to evaluate the accuracy and robustness of current models for text-to-image generation. The models we used include (1) GAN-based models pretrained on MSCOCO: StackGAN (Zhang et al., 2017) which generates high-resolution images by stacking two GANs and DF-GAN (Tao et al., 2020) which is a new one-stage text-to-image backbone that enhances the text-image semantic consistency with a target-aware discriminator; (2) DALL-Es: Since the pretrained checkpoints of original DALL-E (Ramesh et al., 2021) have not been released, we experiment with two open-sourced implementations of DALL-E by (Dayma et al., 2021), i.e., DALLE-mini [1] and DALLE-mega [2]. During training, a VQ-GAN (Esser et al., 2021) model pretrained on ImageNet is used as the image encoder. A BART trans-

---

[1] DALLE-mini: `https://huggingface.co/dalle-mini/dalle-mini`
[2] DALLE-mega: `https://huggingface.co/dalle-mini/dalle-mega`

Table 1: Evaluation of Text-to-Image Synthesis models with CLIP score and our proposed MIS Score by measuring MIS-subject, MIS-object, and MIS-relation for the corresponding semantic concepts in the input prompts. Results are shown as percentages.

| Method | CLIP↑ | MIS-subject | MIS-object | MIS-relation | MIS-Score↑ |
|---|---|---|---|---|---|
| Reference images | 29.7 | 28.3 | 27.5 | 28.0 | 83.8 |
| StackGAN | 15.4 | 14.2 | 9.1 | 14.6 | 37.9 |
| DF-GAN | 12.6 | 16.8 | 12.6 | 15.8 | 45.2 |
| DALLE-mini | 22.6 | 20.5 | 17.9 | 24.7 | 63.1 |
| DALLE-mega | 23.5 | 22.3 | 21.7 | **25.5** | 69.5 |
| Composable Diffusion | 27.9 | 24.3 | **24.8** | 22.2 | **71.3** |
| Stable Diffusion | **28.6** | **24.5** | 21.6 | 24.7 | 70.8 |

former (Lewis et al., 2019) is trained on 15M image-text pairs from Conceptual Captions (Sharma et al., 2018; Changpinyo et al., 2021) and OpenAI subset of YFCC100M (Thomee et al., 2016); (3) Diffusion-based models: Composable Diffusion (Liu et al., 2022) which introduces a compositional operators with multiple diffusion models, and Stable Diffusion (Rombach et al., 2022) which is a latent text-to-image diffusion model using a frozen CLIP ViT-L/14 text encoder to condition the model on text prompts and trained a subset of LAION-5B (Schuhmann et al., 2022). We adopt the open-sourced code in colab[3] for experiments.

**Semantic concept parsing.** While such models are highly flexible, they struggle to understand the composition of certain concepts, such as confusing the attributes of different objects or relations between objects. In our experiments, sentence parsing is first applied to parse or decompose the prompt sentence into different semantic concepts. In linguistics, parsing means to break down a sentence into parts to understand the sentence in a comprehensive manner. In our experiments, we selected the three most common terms in linguistic typology: *subject*, *object*, and the *relation* between the former two terms. The weight parameter is set as 1/3 to calculate the mean average value on all terms.

**Visual components validation.** Our model is pretrained on RefCOCO (Yu et al., 2016) for visual grounding. The feature encoder is composed of CNN, transformer encoder layers and BERT. We use ResNet-101 as the CNN backbone followed by 6 transformer encoder layers in the visual feature extraction branch. These layers are initialized with the corresponding weights of DETR (Carion et al., 2020). The textual embedding encoder is initialized with BERT (Devlin et al., 2018).

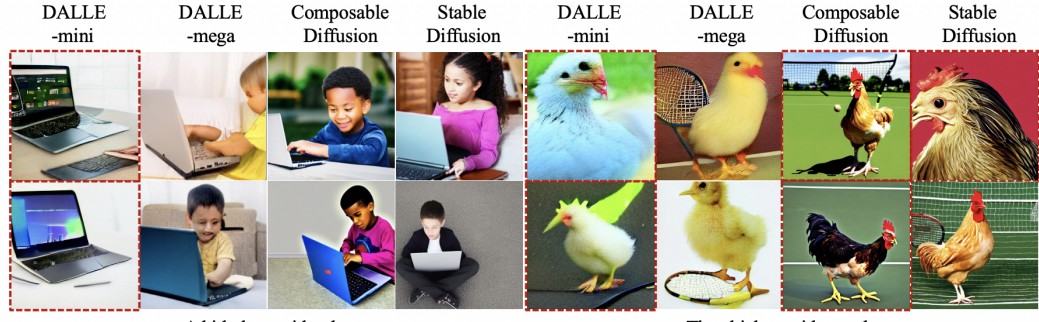

Figure 4: Visualization of text-to-image generation by DALLE-mini, DALLE-mega, Composable Diffusion, and Stable Diffusion. Images with red dotted border represent inaccurate generations of specific semantic concepts.

## 3.2 EVALUATION OF MANIFESTATION: DEVILS IN THE DETAILS

Though MSCOCO (Lin et al., 2014) is a valuable benchmark, it is increasingly clear that it has a limited spectrum of prompts that lack enough insight into differences between generation models.

---

[3] https://colab.research.google.com/github/energy-based-model/
Compositional-Visual-Generation-with-Composable-Diffusion-Models-PyTorch

Table 2: Evaluation on text-to-generation results re-ranked with MIS Score.

| Method | CLIP↑ | MIS-sub. | MIS-obj. | MIS-rel. | MIS-Score↑ | SOA↑ |
|---|---|---|---|---|---|---|
| DALLE-mini | 17.2 | 20.5 | 18.9 | 24.7 | 64.1 | 58.2 |
| DALLE-mega | 20.6 | 23.2 | 22.3 | **25.5** | 71.0 | 63.6 |
| Composable Diffusion | 25.3 | 24.3 | **24.8** | 22.2 | **71.3** | **65.4** |
| Stable Diffusion | **27.8** | **24.5** | 21.6 | 22.7 | 69.8 | 64.1 |

Recent works propose to evaluate models on smaller but more diverse natural language descriptions to systematically evaluate visual generation capability and the underlying biases. Motivated similarly, we introduce GroundBench, a comprehensive set of prompts that support the evaluation and comparison of text-to-image models. Unlike previous methods, GroundBench is sourced from Ref-COCO (Yu et al., 2016) with paired images and texts, and contains diverse prompts to test different capabilities of models such as the ability to faithfully render various objects, spatial relations and some unpopular interactions between objects. We also filter out prompts with a length larger than 15 words and prompts with rare words. Hence, it is easier to use and more flexible to apply in different experimental settings. GroundBench comprises 2368 prompts in total, satisfying the need for a diverse and comprehensive dataset. More details are available on our project webpage [4].

**Quantitative results.** The evaluation of the recent text-to-image models is reported with quantitative results in Table 1. We compare our proposed MIS-Score with CLIP score (Radford et al., 2021), which calculates the cosine similarity of the global features of the image and text. Unlike the CLIP score, MIS-Score measures the distances between the fine-grained inspection terms (i.e., "subject", "object", "relation") and the grounded regional visual features. It is worth noting that our work focuses on measuring the cross-modal semantic consistency rather than the reality of image quality. Some visualization examples are shown in Fig. 4. With MIS-Score, it is easy to obtain more insights into individual semantic concepts: (i) Stable Diffusion generates better results aligned with the subject term in the prompt; (ii) Composable Diffusion performs better on generating the object term than other models; (iii) The diffusion-based models under-perform DALLE-mega in aligning the generated visual components with the relationships.

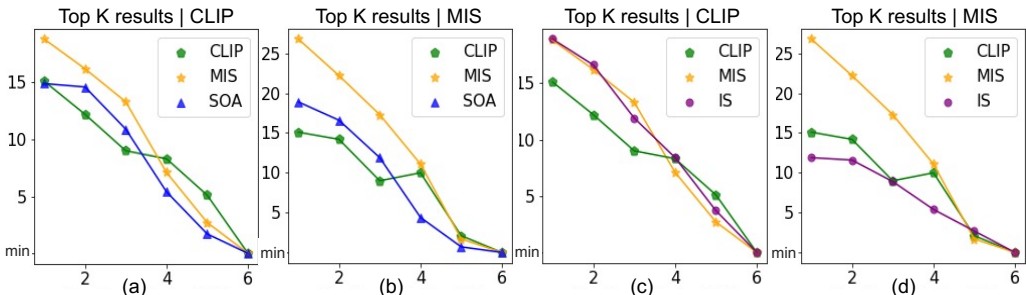

Figure 5: Different metrics evaluated on results ranked by CLIP score in (a) and (c) and results ranked by MIS-Score in (b) and (d). Comparison with SOA score in (a-b), with IS in (c-d).

**Better semantic-consistent results with MIS-Score.** Most text-to-image works report metrics of image quality, *e.g.*, FID and IS, which are not informative enough. CLIP score measures the text-image similarity which has limitations in verifying fine-grained semantics. We show results generated by DALLE-mega ranked by CLIP and MIS-Score in Fig. 6. CLIP focuses more on matching images with the "sandwich" while MIS-Score better balances the verification on another semantic concept of the "plate". In Table 2, we report the evaluation of results re-ranked by MIS-Score. SOA (Hinz et al., 2020) measures the accuracy of objects across modalities. Compared to CLIP, MIS better measures the consistency of semantic concepts and provides more insights in the fine-grained alignment. Additionally, we compare metrics CLIP, MIS with SOA on the top $K$ results ranked by CLIP and MIS. For results ranked by CLIP in Fig. 5(a), MIS also gives a similar ranking and more distinguishable scores. And in Fig. 5(b), CLIP is more fluctuant on results ranked by MIS.

---

[4] https://anonymous.4open.science/r/MIS-Score-D928

Table 3: Robustness testing on DALL-Es and Diffusion-based models.

| Method | Original prompt | Mutate-sub. | Mutate-obj. | Mutate-rel. |
|---|---|---|---|---|
| DALLE-mini | 63.1 | 58.2 | 52.6 | 47.8 |
| DALLE-mega | 69.5 | 65.3 | 59.9 | 54.6 |
| Composable Diffusion | 71.3 | 66.7 | 64.5 | 64.2 |
| Stable Diffusion | 70.8 | 67.2 | 64.1 | 59.4 |

In accordance with SOA, MIS better reflects the alignment on semantic accuracy. Given results in Fig. 5(c-d), we compare CLIP, MIS with IS. It is shown that the results ranked by MIS-Score are less entangled with image quality.

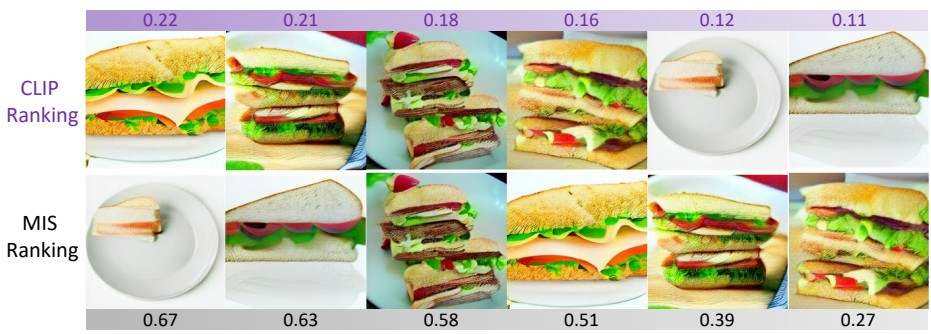

Figure 6: Images generated given the prompt *"A sandwich placed on a plate"*. The first row shows the ranking results by the CLIP score. The second row shows the results re-ranked by MIS-Score.

## 3.3 ROBUSTNESS TESTING WITH REFERENTIAL TRANSFORM

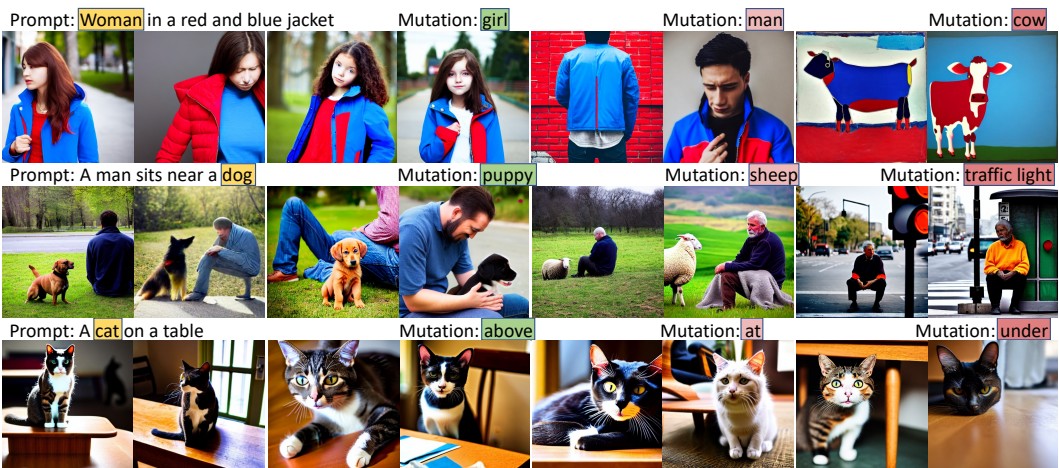

Figure 7: We show the generation by Stable Diffusion with different mutations of semantic concepts on different prompts: the first row mutates the *subject* term, the second row shows mutation on the *object* term, and the last row mutates the *relation* between the *subject* term and the *object* term.

Most multi-modal image synthesis models are trained or using pretrained weights on large-scale datasets. The biases in the data collection are rooted in-depth. The effect of task-specific training on biases is an interesting open question. With referential transform, we operate three types of mutations for testing and results are listed in Table 3. Overall, the results are consistent with prior analysis (Cho et al., 2022; Rombach et al., 2022) where the performances of both DALLE-based and diffusion-based models drop when generating images with mutated prompts. We also have

additional evaluation results on different types of mutations on subject, object and relation terms. They are not identical based on which biases are discovered.

In Table 3, it is discovered that Stable Diffusion performs the best for mutations on the *subject* term. Composable Diffusion is more robust to different mutations on the *relation* term. Besides, comparing the results of DALLE-mini and DALLE-mega, it can be observed that DALLE-mega which is trained on more data is more robust to the referential transforms. More visualized examples are given in Fig. 7. The results indicate that the model tend to struggle for generating long-tailed semantic concepts and rare combinations of different concepts.

## 4 RELATED WORK

**Multi-modal Image Synthesis.** The goal of multi-modal image synthesis is to create realistic images with natural textures, given cross-modal guidance, *e.g.*, using text descriptions as a flexible and natural way to express visual concepts. The prosperity of large-scale Generative Adversarial Networks (GANs) (Brock et al., 2018; Karras et al., 2019; 2020; 2021) significantly advances research on multi-modal image synthesis. By leveraging contrastive language-image pre-training (CLIP) (Radford et al., 2021) or GAN inversion techniques (Xia et al., 2022), pre-trained GANs become applicable to achieve text-driven image synthesis and editing tasks. Recently, the power of handling input from multiple modalities has led to the popularity of Transformer models (Vaswani et al., 2017). Significant progress has been made in multi-modal image synthesis. For example, DALL-E (Ramesh et al., 2021; 2022) trained a large-scale auto-regressive Transformer on a large amount of image-text pairs to produce a high-fidelity generative model through text prompts. For another example, Stable Diffusion (Rombach et al., 2022) adopted latent diffusion models to achieve favorable results across a wide range of multi-modal image synthesis tasks. Although high-fidelity image synthesis is significantly advanced by those efforts, the fine-grained semantic alignment between the text and the generated images is seldom quantitatively investigated.

**Evaluation and Limitation.** The rapid advance in multi-modal image synthesis offers unprecedented generation realism and editing possibilities, which have influenced and will continue to influence our society. The establishment of an accurate, reliable and systematic evaluation framework is highly necessary to evaluate multi-modal image synthesis models and direct future directions. However, evaluating the quality of the image generative models has been proven to be challenging as demonstrated in (Theis et al., 2015). In most of the multi-modal image synthesis methods (Ramesh et al., 2021; 2022; Rombach et al., 2022), image quality and text-image alignment are the main factors considered in the evaluation process. Commonly used evaluation metrics are Inception Score (IS) (Salimans et al., 2016) and the Frechet Inception Distance (FID) (Heusel et al., 2017) for image fidelity, and CLIP score (Radford et al., 2021) for text-image alignment. Because none of the existing measures are perfect, it is usual to report many metrics, each of which assesses a certain aspect. The performance assessment is even more challenging in the text-to-image synthesis task due to the multi-modal complexity of text and image, which motivates us to develop a new evaluation metric to compare text-to-image alignment fairly and confidently.

## 5 CONCLUSION

Although a line of methods has been proposed to advance the multi-modal image synthesis tasks, there have been comparatively fewer works investigating metrics for better evaluation. Most existing metrics concentrate on measuring the image quality but are not informative enough in the find-grained semantic space, where different modalities should be well-aligned. In this work, we propose a new evaluation metric for better understanding and measuring the fine-grained semantic consistency across different modalities. We believe this is a critical avenue for future research and advancement. It is demonstrated in our experiments that existing state-of-the-art text-to-image models exhibit limitations when generating images depicting certain textual descriptions. Besides, we also propose referential transform for testing the robustness of models. It is discovered in our assessment that existing models underperform when synthesizing semantic concepts in the context. They may fail when the mutation goes against certain phrases or commonsense. Developing methods to mitigate the biases and improve the robustness will be an interesting future research direction.

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
