# OpenReview forum: "Probing into the Fine-grained Manifestation in Multi-modal Image Synthesis"
_ICLR.cc/2023/Conference — Submitted to ICLR 2023_

### Official Review · Reviewer_1VeC · 2022-10-23

**Confidence:** 4
**Clarity, Quality, Novelty And Reproducibility:** Clarity, Quality, Novelty, and Reprod…
**Correctness:** 3
**Technical Novelty And Significance:** 3
**Empirical Novelty And Significance:** 3
**Recommendation:** 6

**Strength And Weaknesses:**

Strength:
1, The first time to propose a new approach for measuring the fine-grained semantic consistency for the multi-modal image synthesis tasks.
2, A robustness testing technique is designed and implemented to evaluate the robustness of text-to-image generation models.
3, Comparisions between different methods are interesting.
Weaknesses:
1, How to calculate weight wi in eq3?
2, Can you calculate your metric directly on the benchmark dataset to check the annotation quality of the caption?
3, Why Stable Diffusion is not the best model here?  It seems that they use more data.
4, Do you still need a pretrained CLIP to calculate your score?
5, How can you locate the object region?

**Summary Of The Paper:**

1, Present a new method to evaluate the cross-modal consistency by inspecting the decomposed semantic concepts.
2, Introduce a new metric, called MIS-Score, which is designed to measure the fine-grained semantic alignment between a prompt and its generation quantitatively.
3, Developed an automated robustness testing technique with referential transforms to test and measure the robustness of multi-modal synthesis models.

**Summary Of The Review:**

A new metric is designed for measuring the fine-grained semantic consistency for the multi-modal image synthesis tasks. The measurement for multi-modal image synthesis tasks are important.

---

> ### Author Response · Authors · 2022-11-10
> **Our model measures the fine-grained semantic consistency by learning the visual grounding task**
>
> 1. The weights in Eq3 are hyper-parameters that can be customized. In the experiments, they are set to 1/n, n is the number of semantic concepts extracted from the input prompt as mentioned in Sec3.1.
> 2. Yes, the metric scores on the benchmark dataset with text-image pairs can be found in the first row (reference images) in Table 1.
> 3. In our observation, it is true that the Stable-Diffusion model trained on more data can generate images with good quality (fidelity and diversity). However, since MIS Score measures the fine-grained semantic consistency between the input text and output image, it is discovered that Composable-Diffusion does a better job in manifesting the object terms than Stable-Diffusion. It also proves that our metric can verify the consistency of fine-grained parts than CLIP score which simply matches the global embedding of image-text pairs.
> 4. No, we don’t need a pretrained CLIP. Our model adopts the most popular backbones: ResNet-101 as the image encoder and BERT as the text encoder and is trained on the RefCOCO dataset for the grounding task. The proposed score is calculated with features learned in the intermediate layers.
> 5. In order to measure the fine-grained semantic consistency, our model is designed to locate each semantic concept by learning the grounding task: given a sentence or a phrase, the model is trained to locate the most relevant region in the image.

---

### Official Review · Reviewer_jHfC · 2022-10-25

**Confidence:** 4
**Correctness:** 2
**Technical Novelty And Significance:** 2
**Empirical Novelty And Significance:** Not applicable
**Recommendation:** 3

**Clarity, Quality, Novelty And Reproducibility:**

The novelty of proposed metric might be limited, but authors provide sufficient information to reproduce the proposed method.

**Strength And Weaknesses:**

Strength:

1. The paper is well written and easy to follow.

2. Authors further consider the fine-grained semantic consistency between the given text and synthetic image to completely evaluate the correlation between them.

Weaknesses:
1. Basically, the proposed metric mainly focuses on measuring the semantic consistency between the given text and the generated image, which is different from IS and FID. So some statements shown in introduction, e.g., "These metrics work well for the generation from simple prompts, e.g., description of a single object. However, for prompts with multiple objects and additional context information, simply adopting these metrics is insufficient and may lead to inaccurate or inconsistent results. " might not be completely accurate. Also, the R-precision and CLIP score can also be adopted to measure the correlation between multiple object by adopting a similar detector used in the propose method.

2. Although the proposed MIS-Score further considers the fine-grained correlation between the text and image, it mainly depends on a parser to find semantic concepts and a detector, so its performance might be affected by both components.

3. For equation 2, could authors give more details to explain why it can be used to measure the correlation?

**Summary Of The Paper:**

The paper proposes a new metric to evaluate the image quality for multi-modal image synthesis task,  which relies on a parser to find different semantic concepts and a detector to find the corresponding objects and attributes in the image. Besides, a robustness testing technique is proposed to evaluate the robustness of a generative model.

**Summary Of The Review:**

Please see above weaknesses, and I am happy to change my rating based on authors' responses.

---

> ### Author Response · Authors · 2022-11-15
> **Our work proposes an automatic evaluation method and easy-to-apply metric for measuring the multi-modal semantic consistency**
>
> 1. The motivation and advantages of our work are:
> - Most current works apply CLIP to measure the cross-modal semantic consistency by calculating the similarity of the global embeddings of image-text pairs. In our observation, it struggles to evaluate the generation of long prompts composed of multiple entities. This has also been discussed in [review and diffusion]. To date, there is no metric considering the fine-grained semantic consistency.
> - Fine-grained consistency can be realized by combining detection/classification models and metrics like R-precision. However, the accuracy of detection models is limited and cannot capture detailed attributes like colors and shapes. Instead, our model can locate a text phrase describing an object or the relationship in the image. Thus, our method is more flexible and comprehensive by first decomposing the prompt and matching each semantic phrase to the corresponding image region.
>
> 2. Thanks for this question. Our method is built upon the well-developed parsing and grounding techniques:
> - First, the parser has been widely used in language-related tasks. Our method is not sensitive to the accuracy of the parsing since we focus on evaluating the subject/object/relationship terms by recognizing the part of speech of noun/prep words.
> - Secondly, the performance of our location module is promised by learning the grounding task: given a language phrase, the model will locate the corresponding region in the image. Our benchmark is extended from Ref-COCO which is a subset of MS COCO. The model is trained on the selected text-phrase/image-region pairs. We will add more details of model training and evaluation in the manuscript.
>
> 3. The mathematics behind Eq2 is similar to the implementation of cosine similarity which measures the cosine of the angle between two vectors projected in a multi-dimensional space. The two embeddings have been normalized before Eq2. Furthermore, we applied exponential smoothing which is easy to learn and gives more discriminative values of different observations.

---

### Official Review · Reviewer_nMLm · 2022-10-28

**Confidence:** 3
**Correctness:** 2
**Technical Novelty And Significance:** 2
**Empirical Novelty And Significance:** 1
**Recommendation:** 3

**Clarity, Quality, Novelty And Reproducibility:**

The introduction of the method is limited, and it is difficult to capture the details of the method.

**Strength And Weaknesses:**

+The author proposes a new metric for evaluating multi-mode image synthesis methods
+The author used a large number of experiments to verify

-Recently, there are many meaningful works for multi-mode image synthesis. I don't know why the author has not conducted more detailed research in the Introduction section. Instead, follow the work of 2020 and 2021.

Dalmaz O, Yurt M, Çukur T. ResViT: residual vision transformers for multimodal medical image synthesis[J]. IEEE Transactions on Medical Imaging, 2022, 41(10): 2598-2614.
Isaac-Medina B K S, Bhowmik N, Willcocks C G, et al. Cross-Modal Image Synthesis Within Dual-Energy X-Ray Security Imagery[C]//Proceedings of the IEEE/CVF Conference on Computer Vision and Pattern Recognition. 2022: 333-341.
Wu G, Chen X, Shi Z, et al. Convolutional neural network with coarse-to-fine resolution fusion and residual learning structures for cross-modality image synthesis[J]. Biomedical Signal Processing and Control, 2022, 71: 103199.
Lv Z, Li X, Niu Z, et al. Semantic-shape Adaptive Feature Modulation for Semantic Image Synthesis[C]//Proceedings of the IEEE/CVF Conference on Computer Vision and Pattern Recognition. 2022: 11214-11223.
Tan Z, Chu Q, Chai M, et al. Semantic Probability Distribution Modeling for Diverse Semantic Image Synthesis[J]. IEEE Transactions on Pattern Analysis and Machine Intelligence, 2022.

-In denying that Measure is an important metric, as Section 7 of [1], I hope that the author can further compare the advantages of the metric and have more proof of the metrics.

[1] Zhou R, Jiang C, Xu Q. A survey on generative adversarial network-based text-to-image synthesis[J]. Neurocomputing, 2021, 451: 316-336.

-In the introduce section, the author introduces MIS-Score, but it is difficult capture the first contribution point and the third contribution point by reading this section.

-The introduction of the method is limited, and it is difficult to capture the details of the method. The author's description of the implementation framework is incomplete and it is difficult to understand the proposed framework.


**Summary Of The Paper:**

The author proposes a new metric for evaluating the multi-mode image synthesis method, and has tested new metrics. However, method details are limited. In addition, the author's contribution point 1,3 is difficult to capture in the introduction. Besides, the papers investigated are old.

**Summary Of The Review:**

The research topic is worth recommending, but the author's description of the method needs to be strengthened.

---

> ### Author Response · Authors · 2022-11-10
> **Our method provides a novel view of evaluating the fine-grained semantic consistency**
>
> 1. In this work, we compared three typical types of text-to-image models: GAN-based models, DALL-E, and diffusion-models. Stable Diffusion is accepted by CVPR22 and Composable Diffusion by ECCV22 which are the SoTA methods. This paper does not focus on choosing the best model. Instead, we develop a new metric to evaluate the fine-grained consistency. The reason for not adopting other works in the experiment is that most recent works are not open-sourced yet and are resource-consuming to train on large-scale datasets. In our future work, we will include more approaches in the introduction and evaluation.
>
> 2. Thanks for this advice. In this work, we provide a novel view of evaluating the consistency of the rich semantic details across the input prompts and the generated images. We propose a grounding-based method that can be a complementary measurement for current metrics. The main advantages of MIS Score are as follows:
> - Fine-grained consistency measurement: MIS Score measures the matching similarity between each text phrase with its corresponding visual region. While the CLIP score only considers the global similarity between the sentence embedding and the visual features.
> - Better evaluation of the generated content: With a structured understanding of both the text and image contents, MIS Score can better evaluate the semantic consistency and is less affected by the quality of generated images.
>
> 3. In the introduction section, we mainly elaborate on the background and motivation of this work. Due to the page limitation, more results and discussion can be found in the experiment section.
> - As our first contribution, we propose to measure the fine-grained semantic consistency by first decomposing the input text into semantic phrases and grounding the local visual region of each phrase. MIS Score is designed to calculate the similarity taking account of the fine-grained cross-modal semantic matching.
> - As for the thirst contribution, a new mechanism is proposed to test the robustness of different generation models by replacing the semantic phrase with another phrase that is inspired by referential transparency. The robustness testing is needed since there are many biases in the training datasets where certain types of objects appear in a set phrase. Current models often fail for generating rare objects or combinations as shown in Fig.7.
>
> 4. The proposed method is composed of three main modules: text encoder(BERT), image encoder(ResNet101), and cross-model verification module. We adopted the most popular models as encoders to leverage the pre-trained weights on large-scale datasets. A detailed description of the verification module can be found in Section2.2. The proposed framework is illustrated in Fig.2: With the input prompt and generated image from Fig.2(a), we first parse the sentence into different parts and locate each phrase in the image region in Fig.2(b). Thereafter, MIS Score calculates the similarity of each fine-grained text phrase and the corresponding image region features.

---

### Decision · Program_Chairs · 2023-01-20

**Decision:**

Reject

**Justification For Why Not Higher Score:**

Reviewers raised a number of concerns related to novelty, evaluation, and lack of implementation details. These concerns reasonably persist even after accounting for the authors' response.

**Justification For Why Not Lower Score:**

N/A

**Metareview: Summary, Strengths And Weaknesses:**

This paper proposes a new metric to evaluate image quality in multi-modal image synthesis (i.e. text-to-image synthesis). It is specifically interested in assessing how well the generated imagery respects the input text prompt, and it claims to do so at a finer level of granularity than previously-available metrics.

Strengths: reviewers found the paper well-written and easy to follow. They also appreciated the extensive experimental evaluation.

Weaknesses: all reviewers expressed a concern about a lack of novelty. There already exist methods for assessing consistency between text prompts and generated images (e.g. CLIP score, R-precision), and the reviewers were not convinced that the method proposed here offered anything useful beyond those.

Overall, reviewers were not in favor of accepting the paper, given this weakness.

The authors wrote very thorough responses to each of the reviewers, but none of the reviewers responded during the OpenReview discussion period. Thus, I decided that an in-person meeting was warranted.

**Summary Of Ac-Reviewer Meeting:**

Two of the three reviewers attended the meeting.

My goal to was ascertain what these reviewers thought about the paper in light of the authors' responses (since none of them had acknowledged these responses on OpenReview).

Both reviewers were still negative about the paper. One commented that the quality of results that the method could obtain seems critically dependent on how good the parser is that it uses to parse the input prompt. There was no evaluation of how the performance of the entire method degrades as the parser's outputs become imperfect (especially on longer and/or more complex sentences). Perhaps more critically: even when the parser is working well, it doesn't appear to offer any meaningful advantages over a simpler potential baseline of just running an object detector on the image and then checking that the detected objects appear in the input sentence. In other words: while a more detailed parse of the input sentence could *in principle* be used to get at a more fine-grained notion of input/output consistency, the proposed method doesn't really show how specifically to do this.

Another reviewer commented on some of the content of the authors' responses. The authors claimed that they couldn't compare to much prior work b/c of a lack of existing open-source implementations; however, this reviewer found several relevant prior works with available code:

1. Reed S, Akata Z, Yan X, et al. Generative adversarial text to image synthesis[C]//ICLR, 2016: 1060-1069. （https://github.com/soumith/dcgan.torch）

2.Li W, Zhang P, Zhang L, et al. Object-driven text-to-image synthesis via adversarial training[C]//CVPR. 2019: 12174-12182.（https://github.com/jamesli1618/Obj-GAN）

3.Zhang H, Xu T, Li H, et al. Stackgan: Text to photo-realistic image synthesis with stacked generative adversarial networks[C]//ICCV. 2017: 5907-5915. （https://github.com/hanzhanggit/StackGAN）

4.Zhang H, Xu T, Li H, et al. Stackgan++: Realistic image synthesis with stacked generative adversarial networks[J]. IEEE transactions on pattern analysis and machine intelligence, 2018, 41(8): 1947-1962.（https://github.com/hanzhanggit/StackGAN-v2）

5.AttnGAN: Fine-Grained Text to Image Generation with Attentional Generative Adversarial Networks（Xu T, Zhang P, Huang Q, et al. Attngan: Fine-grained text to image generation with attentional generative adversarial networks[C]//CVPR. 2018: 1316-1324.）

6.Photographic Text-to-Image Synthesis with a Hierarchically-nested Adversarial Network（Zhang Z, Xie Y, Yang L. Photographic text-to-image synthesis with a hierarchically-nested adversarial network[C]//CVPR. 2018: 6199-6208.）

7.Li Y, Gan Z, Shen Y, et al. Storygan: A sequential conditional gan for story visualization[C]//Proceedings of the IEEE/CVPR. 2019: 6329-6338.（https://github.com/yitong91/StoryGAN）

Finally: this same reviewer expressed concern about how the authors wrote in their response that they were not able to provide some details about their implementation because of the strict page limit. This seems somewhat suspicious, given that authors are free to provide additional implementation details in the supplemental material.

In light of these discussions, I feel comfortable recommending that the paper be rejected.